# Breaking the trade-off between fast control and long lifetime of a superconducting qubit

S. Kono [1✉], K. Koshino[2], D. Lachance-Quirion[3,4], A. F. van Loo [1], Y. Tabuchi [3], A. Noguchi [5,6] & Y. Nakamura [1,3✉]

The rapid development in designs and fabrication techniques of superconducting qubits has made coherence times of qubits longer. In the future, however, the radiative decay of a qubit into its control line will be a fundamental limitation, imposing a trade-off between fast control and long lifetime of the qubit. Here, we break this trade-off by strongly coupling another superconducting qubit along the control line. This second qubit, which we call "Josephson quantum filter" (JQF), prevents the first qubit from emitting microwave photons and thus suppresses its relaxation, while transmitting large-amplitude control microwave pulses due to the saturation of the quantum filter, enabling fast qubit control. This device functions as an automatic decoupler between a qubit and its control line and could help in the realization of a large-scale superconducting quantum processor by reducing the heating of the qubit environment and the crosstalk between qubits.

[1] Center for Emergent Matter Science (CEMS), RIKEN, Wako, Saitama 351-0198, Japan. [2] College of Liberal Arts and Sciences, Tokyo Medical and Dental University, Ichikawa, Chiba 272-0827, Japan. [3] Research Center for Advanced Science and Technology (RCAST), The University of Tokyo, Meguro-ku, Tokyo 153-8904, Japan. [4] Institut Quantique and Département de Physique, Université de Sherbrooke, Sherbrooke, QC J1K 2R1, Canada. [5] Komaba Institute for Science (KIS), The University of Tokyo, Meguro-ku, Tokyo 153-8902, Japan. [6] PRESTO, Japan Science and Technology Agency, Kawaguchi-shi, Saitama 332-0012, Japan. ✉email: shingo.kono@riken.jp; yasunobu@ap.t.u-tokyo.ac.jp

Single-qubit gates are an essential element for any quantum protocol based on qubits[1]. They are typically implemented by applying an electromagnetic field in resonance with the energy difference between two levels, inducing Rabi oscillations[2–4]. The qubit has to be coupled to at least one one-dimensional continuous mode to have an external control. Although a larger coupling strength to the control degree of freedom achieves a faster gate operation for a given drive amplitude, it also increases the radiative decay of the qubit into the continuous mode. Conversely, suppressing this radiative decay by reducing the coupling strength leads to slower qubit control. This is a fundamental trade-off between fast control and long lifetime of a qubit, which originates from the fluctuation–dissipation theorem[5].

Superconducting qubits are a promising candidate for a large-scale quantum processor[6]. The ceaseless developments in designs and fabrication techniques have been extending the coherence times of theses qubits[7]. The radiative decay of a superconducting qubit to its control line can no longer be dismissed in devices with state-of-the-art coherence times. The trade-off in qubit control has so far been dealt with by designing a weak coupling to the control line and applying a strong microwave drive field for compensation[8]. However, further improvements in the coherence time of superconducting qubits would require even weaker coupling to the control line, leading to an increase in the microwave power needed to control the qubits. This will be problematic for large-scale superconducting quantum circuits due to heating of the qubit cryogenic environment[8–11] and the output power level of the control electronics[12,13]. Furthermore, the demand for a strong microwave drive field may increase crosstalk to non-targeted qubits in the vicinity[14].

Here, we experimentally demonstrate the suppression of the radiative decay of a "data" qubit to its control line without sacrificing the gate speed by using an ancillary qubit that acts as a nonlinear filter. We name this filter a Josephson quantum filter (JQF)[15]. As shown in Fig. 1a, on one hand, the JQF reflects single photons emitted from the data qubit, suppressing the radiative decay to the control line. On the other hand, when a large-amplitude control field is applied (Fig. 1b), the JQF saturates and becomes transparent, enabling fast Rabi oscillations of the data qubit. The working principle is in contrast to that of a Purcell filter, which utilizes the frequency difference between a qubit and a readout resonator to realize both fast readout and long lifetime

of the qubit[16–18]. The Purcell filter circuit is not suitable, however, for a case where the frequencies of the radiative decay and the control signal are identical.

## Results

**Theoretical model.** A system composed of a data qubit and a JQF in a semi-infinite control line is described theoretically by the waveguide-quantum-electrodynamics formalism[19–23]. As shown in Fig. 1c, d, the data qubit is placed at the end of the control line, while the JQF is located a distance $d$ away from the qubit[24,25]. Here, we consider that the JQF frequency $\omega_f$ is set to be identical to the qubit frequency $\omega_q$. The resonant interaction mediated by photons in the control line induces two cooperative effects depending on the distance: a correlated decay and an energy-exchange interaction[20,21]. In the frame rotating at the qubit and JQF frequencies, the master equation of the composite system of the qubit and JQF with a resonant control field is given by

$$\dot{\hat{\rho}} = -\frac{i}{\hbar}\left[\hat{H}_{eff} + \hat{H}_{drive},\ \hat{\rho}\right] + \sum_{i,j=q,f} \gamma_{ex}^{ij}\ \mathcal{D}(\hat{\sigma}_i, \hat{\sigma}_j)\hat{\rho}, \quad (1)$$

where $\hat{\sigma}_i$ ($i = $ q, f) is the respective lowering operator of the qubit and the JQF, and $\mathcal{D}(\hat{A}, \hat{B})\rho = \hat{B}\hat{\rho}\hat{A}^\dagger - (\hat{A}^\dagger\hat{B}\hat{\rho} + \hat{\rho}\hat{A}^\dagger\hat{B})/2$ is a superoperator describing the correlated decay. The correlated decay terms are described with the individual decay rates of $\gamma_{ex}^{qq} = \gamma_{ex}^q$ and $\gamma_{ex}^{ff} = \gamma_{ex}^f\cos^2(2\pi d/\lambda_q)$, and the correlated decay rates of $\gamma_{ex}^{qf} = \gamma_{ex}^{fq} = \sqrt{\gamma_{ex}^q\gamma_{ex}^f}\cos(2\pi d/\lambda_q)$, where $\gamma_{ex}^i$ ($i = $ q, f) is the respective external coupling rate and $\lambda_q$ is the qubit wavelength. The effective energy-exchange interaction is described as $\hat{H}_{eff} = \hbar J(\hat{\sigma}_q^\dagger\hat{\sigma}_f + \hat{\sigma}_q\hat{\sigma}_f^\dagger)$, where $J = \sqrt{\gamma_{ex}^q\gamma_{ex}^f}\sin(2\pi d/\lambda_q)/2$ is the coupling strength. Moreover, the drive Hamiltonian is given by $\hat{H}_{drive} = \hbar\Omega_q\ \hat{\sigma}_x^q/2 + \hbar\Omega_f\cos(2\pi d/\lambda_q)\hat{\sigma}_x^f/2$, where $\Omega_i = 2\sqrt{\gamma_{ex}^i\dot{n}}$ is the Rabi frequency with a control photon flux of $\dot{n}$ and $\hat{\sigma}_s^i$ is the Pauli $s$ operator ($s = x, y, z$).

Here, we explain the working principle of the JQF based on the master Eq. (1). The seemingly contradictory goal is that the qubit is isolated from the JQF while employing the subradiance effect protecting the qubit from decaying. We achieve this by preparing a system where the qubit and the JQF are coupled with the strongly asymmetric external coupling rates to the control line, as

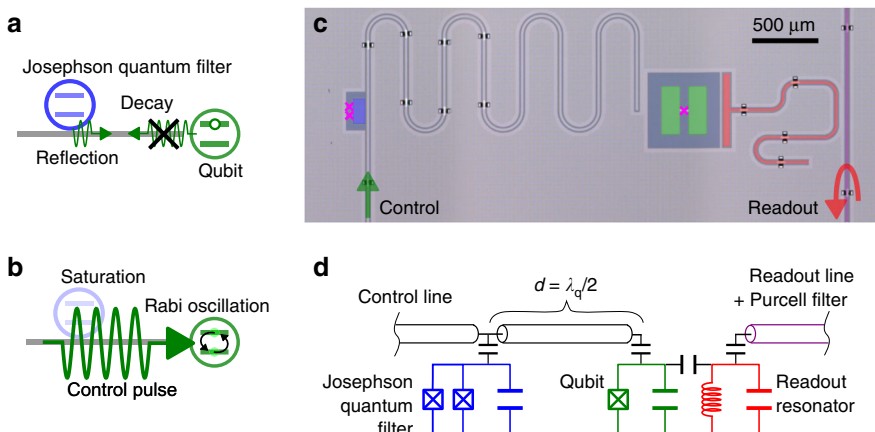

**Fig. 1 Josephson quantum filter (JQF). a, b** Concept of the JQF. The JQF reflects single photons emitted from the qubit, suppressing the qubit radiative decay. In contrast, when a large-amplitude control pulse is applied to the qubit, the JQF saturates and transmits the pulse, enabling fast qubit control. **c, d** False-colored optical image and equivalent circuit of the fabricated superconducting circuit. A fixed-frequency transmon qubit acting as the data qubit (green) is connected to a control coplanar-waveguide with an ancillary tunable-frequency transmon qubit with a SQUID acting as the JQF (blue). The data qubit is coupled to a resonator (red) for fast dispersive readout via a readout line with a Purcell filter (purple). Air-bridges are fabricated on the waveguides to suppress spurious modes. The Josephson junctions are indicated by magenta crosses.

follows. To avoid the qubit hybridizing with the JQF ($J = 0$) and to maximize the correlated decay ($|\gamma_{ex}^{qf}| = |\gamma_{ex}^{fq}| = \sqrt{\gamma_{ex}^{q}\gamma_{ex}^{f}}$), the distance $d$ is set to half the qubit wavelength ($d = \lambda_q/2$). Therefore, the correlated decays are described as the individual decays of a bright mode [$\hat{\sigma}_B = \mathcal{N}(-\sqrt{\gamma_{ex}^{q}}\,\hat{\sigma}_q + \sqrt{\gamma_{ex}^{f}}\,\hat{\sigma}_f)$] with decay rate $\gamma_{ex}^{B} = \gamma_{ex}^{q} + \gamma_{ex}^{f}$ and a dark mode [$\hat{\sigma}_D = \mathcal{N}(\sqrt{\gamma_{ex}^{f}}\,\hat{\sigma}_q + \sqrt{\gamma_{ex}^{q}}\,\hat{\sigma}_f)$] with decay rate $\gamma_{ex}^{D} = 0$, where $\mathcal{N}$ is the normalization factor[21]. By engineering the system such that $\gamma_{ex}^{f} \gg \gamma_{ex}^{q}$, the excited state of the qubit is close to the dark state ($\hat{\sigma}_q \approx \hat{\sigma}_D$), suppressing its radiative decay. In the limit of $\gamma_{ex}^{f} \gg \gamma_{ex}^{q}$, the master Eq. (1) can be approximated as

$$\dot{\hat{\rho}} = -\frac{i}{\hbar}\left[\hat{H}_{drive},\ \hat{\rho}\right] + \mathcal{D}\left(\sqrt{\gamma_{ex}^{f}}\,\hat{\sigma}_f - \sqrt{\gamma_{ex}^{q}}\,(1+\hat{\sigma}_z^{f})\hat{\sigma}_q\right)\hat{\rho}, \quad (2)$$

with the effective drive Hamiltonian originating from the correlated decay:

$$\hat{H}_{drive} = -\frac{\hbar\Omega_f}{2}\hat{\sigma}_x^{f} + \frac{\hbar\Omega_q}{2}(1+\hat{\sigma}_z^{f})\,\hat{\sigma}_x^{q}, \quad (3)$$

where $\mathcal{D}(\hat{A})\hat{\rho} = \hat{A}\hat{\rho}\hat{A}^{\dagger} - (\hat{A}^{\dagger}\hat{A}\hat{\rho} + \hat{\rho}\hat{A}^{\dagger}\hat{A})/2$. From this approximative master equation, we find that the qubit is coupled to the control line depending on the state of the JQF. When the control field is absent ($\hat{H}_{drive} = 0$), the JQF is in the ground state ($\hat{\sigma}_z^{f} = -1$), resulting in the complete suppression of the decay term of the qubit. When the control field is applied, on the other hand, the conditional drive term of the drive Hamiltonian ($\propto \hat{\sigma}_z^{f}\hat{\sigma}_x^{q}$) is suppressed by the drive term of the JQF ($\propto \hat{\sigma}_x^{f}$) in the secular approximation, enabling us to individually drive the qubit with the Rabi frequency of $\Omega_q$ (see more details in Supplementary Note 4).

**Experimental device**. In the experiment, a superconducting transmon qubit coupled to a coplanar-waveguide control line is fabricated on a silicon substrate, as shown in Fig. 1c. The resonance frequency, anharmonicity, and external coupling rate of the qubit are $\omega_q/2\pi = 8.002$ GHz, $\alpha_q/2\pi = -398$ MHz, and $\gamma_{ex}^{q}/2\pi = 123$ kHz, respectively. The state of the qubit is dispersively read out via a resonator with resonance frequency $\omega_r/2\pi = 10.156$ GHz, external coupling rate $\kappa_{ex}/2\pi = 2.16$ MHz, and state-dependent dispersive frequency shift $2\chi/2\pi = -1.87$ MHz. A Purcell filter is used to prevent the Purcell decay into the readout line from limiting the qubit relaxation time[18]. An ancillary transmon qubit acting as the JQF is strongly coupled to the control line with external coupling rate $\gamma_{ex}^{f}/2\pi = 112$ MHz. The distance between the qubit and JQF is designed to be half the qubit wavelength ($d \approx 7.5$ mm). The JQF resonance frequency $\omega_f/2\pi$ is tunable between 6.3 and 8.5 GHz with a static magnetic field, which enables us to investigate the behavior of the qubit with and without the JQF in a single device: when the JQF is far detuned from the qubit, the qubit behaves as if the JQF does not exist. The anharmonicity and intrinsic loss rate of the JQF are $\alpha_f/2\pi = -387$ MHz and $\gamma_{in}^{f}/2\pi = 3$ MHz, respectively (see more details in Supplementary Notes 1 and 2).

**Dependence of the JQF reflection spectrum on probe power**. We first characterize the JQF by measuring its reflection spectrum via the control line. In Fig. 2a, b, the amplitude and phase of the reflection spectra are shown for different probe powers. At a smaller probe power of $-146$ dBm, the JQF spectrum is in the over-coupling regime, where the external coupling rate is much larger than the intrinsic loss rate, i.e., $\gamma_{ex}^{f} \gg \gamma_{in}^{f}$. The over-coupling regime of the JQF is required for a perfect reflection of

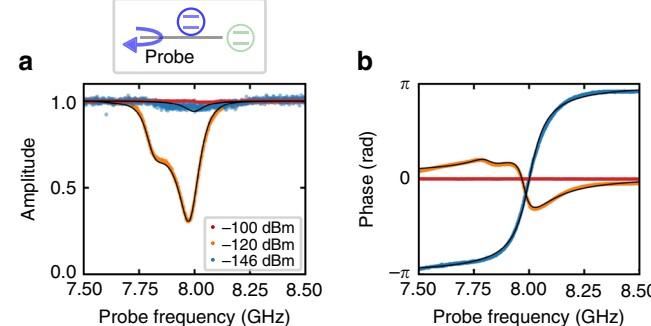

**Fig. 2 Dependence of the JQF reflection spectrum on probe power.**
**a, b** Amplitude and phase of the reflection spectra of the JQF measured via the control line. The dots (blue, orange, red) are the experimental results with different probe powers, and the black lines are the theoretical fits for $-120$ and $-146$ dBm. The probe power of $-120$ dBm corresponds to the single-photon power level for the JQF, defined as $\hbar\omega_f(\gamma_{ex}^{f} + \gamma_{in}^{f})^2/(4\gamma_{ex}^{f})$, which would populate a linear resonator with a single photon on average[30]. Note that the qubit transition is not observed here since the resolution of the probe frequency is larger than the qubit linewidth.

single photons emitted from the qubit[26]. The JQF transition starts to saturate around the single-photon power level ($\approx -120$ dBm). The second dip around 7.8 GHz corresponds to the two-photon transition between the ground and second excited states. At a stronger probe power of $-100$ dBm, the JQF does not affect the reflection coefficient due to it being saturated[26], which is an essential property for allowing the qubit control field to be transmitted through the JQF (Supplementary Note 5).

**Decay times and Rabi frequency of the qubit with and without the JQF**. To study the effect of the JQF on the qubit, we perform time-domain measurements on the qubit with different JQF–qubit detunings. The qubit population in the excited state is obtained from the averaged quadrature amplitude of the readout pulse. Note that the averaged amplitude is normalized by taking into account the thermal population of the qubit for each detuning.

As shown in Fig. 3a, we measure the qubit relaxation with and without the JQF. When the JQF is nearly resonant with the qubit, the qubit shows an exponential decay with a longer relaxation time than without the JQF. The thermal population of the qubit is increased from 2.8 to 16.2%, which is not because the JQF adds a thermal noise to the qubit, but because it decouples the qubit from the control line which has a lower effective temperature than the intrinsic loss channel[27]. The relaxation time ($T_1$) and Hahn-echo coherence time ($T_2^{E}$) of the qubit as a function of the JQF–qubit detuning are shown in Fig. 3b. Both the relaxation and coherence times are enhanced by a factor of about 4 when the JQF is nearly on resonance with the qubit. The enhancement is mainly limited by the intrinsic energy relaxation and thermal population of the qubit. The full frequency bandwidth at half maximum of the enhancement spectrum is found to be about 60 MHz, which roughly coincides with the external coupling rate of the JQF. As this bandwidth corresponds to about 1% of the JQF frequency, it would be possible to implement a frequency-fixed JQF in a future device with the state-of-art Josephson-energy variation[28]. The asymmetry of the enhancement spectrum is explained by the non-ideal JQF–qubit distance, i.e., $d = 0.526\lambda_q$. Interestingly, for this distance, we find in numerical simulations that with a finite detuning of $(\omega_f - \omega_q)/2\pi = 9$ MHz, the decay times of the qubit can reach the level that would be achieved with the ideal distance of $d = 0.5\lambda_q$ (Supplementary Note 9).

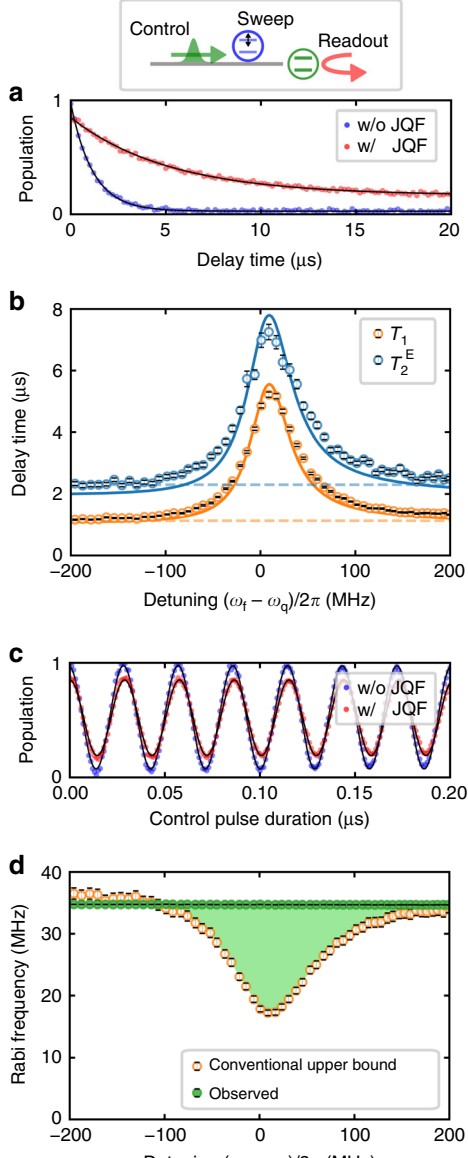

**Fig. 3 Breaking the trade-off between the fast control and long lifetime of the qubit with the JQF.** The red and blue dots are the experimental results with and without the JQF, respectively. The black lines are the theoretical fits. **a** Qubit population in the excited state as a function of the delay time between the $\pi$ pulse and the readout. **b** Decay times as a function of the JQF–qubit detuning. The orange and cyan circles depict the relaxation time ($T_1$) and the Hahn-echo coherence time ($T_2^{\mathrm{E}}$), respectively. The solid lines are the numerically simulated results (see more details in Supplementary Notes 3 and 4). The horizontal dashed lines are those in the absence of the JQF. **c** Qubit population in the excited state as a function of the duration of a square control pulse with Gaussian edges. **d** Rabi frequency as a function of the JQF–qubit detuning. The green circles are the observed Rabi frequency in the presence of the JQF, and the orange circles are the conventional upper bound of the Rabi frequency for the given drive power, which is calculated as $2\sqrt{\dot{n}/T_1}$. The error bars represent standard errors.

Furthermore, Rabi oscillations of the qubit are observed when applying a stationary control field with a photon flux of $\dot{n} = 1.5 \times 10^{10}$ s$^{-1}$, which corresponds to $-101$ dBm. As Fig. 3c shows, the Rabi oscillations are not affected by the presence of the JQF except for the oscillation amplitude, which is decreased due to the thermal excitation caused by the intrinsic loss channel of

the qubit. The observed Rabi frequency as a function of the JQF–qubit detuning is shown with the green circles in Fig. 3d. Due to the saturation of the JQF by the strong control field, the Rabi frequency is found to be constant and does not depend on the detuning.

To further study the trade-off in qubit control, we define the conventional upper bound of the Rabi frequency of the qubit without employing the JQF as $2\sqrt{\dot{n}/T_1}$. This is because a Rabi frequency $\Omega_q$ with a fixed external coupling never exceeds the upper bound, as $\Omega_q = 2\sqrt{\gamma_{\mathrm{ex}}^q \dot{n}} \leq 2\sqrt{\dot{n}/T_1}$. The upper bound can be achieved in a conventional setup only when the intrinsic loss, pure dephasing, and thermal excitation of the qubit are negligible. As shown in Fig. 3d, the observed Rabi frequency with the JQF exceeds this upper bound (indicated by the shadowed area in Fig. 3d), which demonstrates that we break the trade-off in qubit control.

**Controlling the qubit in the presence of the JQF.** To investigate whether the JQF has negative effects on our ability to control the qubit, the Rabi frequency and Rabi decay time of the qubit are measured as a function of the control amplitude (Fig. 4a, b). In the presence of the JQF, in the region where the JQF is not completely saturated ($\Omega_f/\gamma_{\mathrm{ex}}^f \approx 1$), the Rabi frequency is smaller and the decay time is shorter than those without the JQF. However, in the limit of large control amplitudes ($\Omega_f/\gamma_{\mathrm{ex}}^f \gg 1$), the results with and without the JQF become indistinguishable.

In Fig. 4a, b, we compare the experimental results with the numerical ones which we calculate by replacing the transmon JQF with a two-level JQF with the same parameters. Unlike with the transmon JQF, the Rabi decay time of the qubit with the two-level JQF is calculated to be shorter than that in the absence of the JQF, even when the JQF is nearly completely saturated by the control field ($\Omega_f/\gamma_{\mathrm{ex}}^f \gg 1$). This is because the saturated two-level JQF is still coupled to the qubit (see the conditional drive term originating from the correlated decay in Eq. (3)), providing additional decay channels of the qubit through the JQF. The transmon JQF, on the other hand, once excited to its higher levels, becomes decoupled from the qubit due to its anharmonicity and loses the correlated decay, and therefore no longer affects the Rabi oscillations of the qubit. From numerical simulations, we find that the minimum error per qubit Rabi cycle is achieved when the anharmonicity of a transmon JQF almost equals its external coupling rate ($|\alpha_f| \approx \gamma_{\mathrm{ex}}^f$), which agrees with our experimental findings (Supplementary Note 4).

When the qubit is sequentially controlled, the JQF is expected to be saturated during each gate and to not significantly affect the gate fidelity. The average gate error of the Clifford gates on the qubit is measured by using randomized benchmarking[29]. In Fig. 4c, the average gate errors with and without the JQF are shown as a function of the control pulse duration. We confirm that the average gate errors in both cases are close to the coherence limit. The small increase in the gate error with the JQF can be explained by the additional decay of the qubit due to the incomplete saturation of the JQF at the beginning and end of each control pulse. Note that the coherence limit is mainly determined by the external coupling rate of the qubit.

## Discussion
We successfully resolved the trade-off in qubit control by implementing a JQF to the control line of a qubit. We experimentally confirmed that the JQF suppresses the qubit radiative decay, while it does not significantly reduce the Rabi frequency and the gate fidelity of the qubit from those without the JQF. This would also allow us to enhance the qubit–control-line coupling,

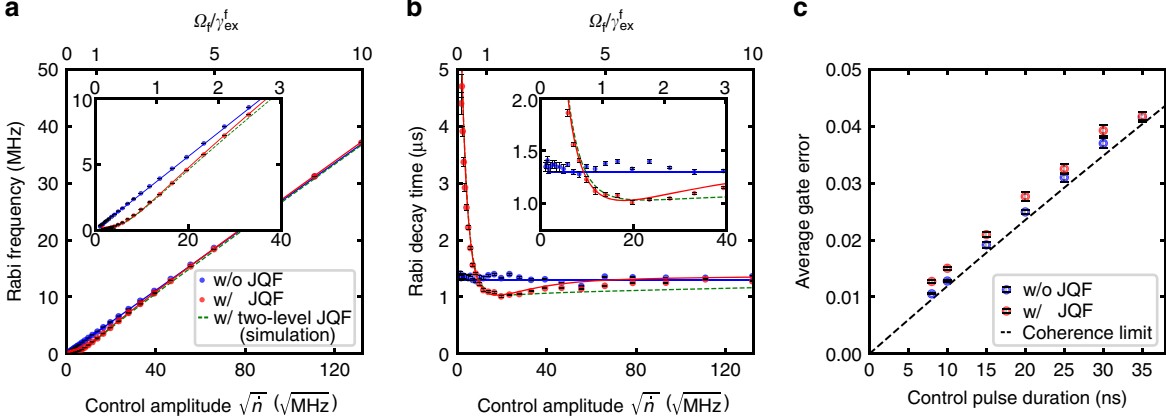

**Fig. 4 Controlling the qubit in the presence of the JQF. a, b** Rabi frequency and Rabi decay time of the qubit as a function of the control amplitude represented as the square root of the photon flux $\sqrt{\dot{n}}$. The top axes show the corresponding Rabi frequency of the JQF $\Omega_{\mathrm{f}}$ normalized by its external decay rate $\gamma_{\mathrm{ex}}^{\mathrm{f}}$. The red and blue data are the experimental results with and without the JQF, respectively. The red and blue solid lines depict numerical results with and without a transmon JQF, whereas the green dashed lines show simulation results using a two-level JQF. The insets are magnifications of the small amplitude region of the main plots. **c** Average gate error of the qubit as a function of the control pulse duration, obtained by randomized benchmarking. A Gaussian pulse is used for the qubit control, and the pulse interval is set to be twice the pulse duration. The black dashed line depicts the coherence limit. The error bars represent standard errors.

and thus to reduce the Rabi drive power, by three orders of magnitude, compared with the weak coupling required for a state-of-the-art qubit with a 1-ms lifetime in the absence of JQF. The device could be useful in the realization of a large-scale superconducting qubit system by reducing the heating of the qubit environment and the crosstalk between qubits. More generally, our experiments show that a nonlinear element acts as a power-dependent variable boundary condition for microwave modes, which can be applied to other types of parametric control, such as two-qubit gates, single-photon generation, or active cooling of quantum systems.

## Methods

**Calibration of the photon flux in the qubit control field.** To quantitatively show the breaking of the trade-off in qubit control, we calibrate the photon flux $\dot{n}$ in the control field. First, we measure the reflection spectrum of the qubit in the absence of the JQF in order to distinguish the external coupling rate of the qubit $\gamma_{\mathrm{ex}}^{\mathrm{q}}$ from the intrinsic loss rate $\gamma_{\mathrm{in}}^{\mathrm{q}}$. By using the fitting results of the reflection spectrum together with the qubit thermal population, the external coupling rate is determined to be $\gamma_{\mathrm{ex}}^{\mathrm{q}}/2\pi = 123$ kHz. Furthermore, given a control amplitude, the corresponding Rabi frequency is obtained by observing the Rabi oscillation of the qubit in the absence of the JQF. Thus, by using the expression of $\Omega_{\mathrm{q}} = 2\sqrt{\gamma_{\mathrm{ex}}^{\mathrm{q}}\dot{n}}$, we calibrate the photon flux $\dot{n}$ for a given control field amplitude. See more details in Supplementary Notes 6–8.

**Conventional upper bound of the qubit Rabi frequency.** We define the upper bound of the Rabi frequency of the qubit controlled in a setup of the fixed external coupling rate as $2\sqrt{\dot{n}/T_1}$. Generally, the total relaxation rate of the qubit is given by $1/T_1 = (2n_{\mathrm{th}}^{\mathrm{q}} + 1)\gamma_{\mathrm{in}}^{\mathrm{q}} + (2n_{\mathrm{th}}^{\mathrm{q,ex}} + 1)\gamma_{\mathrm{ex}}^{\mathrm{q}}$, where $n_{\mathrm{th}}^{\mathrm{q}}$ and $n_{\mathrm{th}}^{\mathrm{q,ex}}$ are the thermal quanta of the intrinsic and external baths, respectively. Therefore, the external coupling rate of the qubit is always smaller than the total relaxation rate, i.e., $\gamma_{\mathrm{ex}}^{\mathrm{q}} \leq 1/T_1$. The external coupling rate of $\gamma_{\mathrm{ex}}^{\mathrm{q}}$ is equal to $1/T_1$ only when $\gamma_{\mathrm{in}}^{\mathrm{q}}$, $n_{\mathrm{th}}^{\mathrm{q,ex}}$, and $n_{\mathrm{th}}^{\mathrm{q}}$ are negligible. Accordingly, the Rabi frequency never exceeds the upper bound, i.e. $\Omega_{\mathrm{q}} = 2\sqrt{\gamma_{\mathrm{ex}}^{\mathrm{q}}\dot{n}} \leq 2\sqrt{\dot{n}/T_1}$. Note that this upper bound can be calculated from $\dot{n}$ and $T_1$, which are obtained by independent experiments.

## Data availability

All the data used in this study are available from the corresponding author upon reasonable request.

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

## Acknowledgements
We acknowledge fruitful discussions with A. Eddins, J. M. Kreikebaum, and K. O'Brien. This work was supported in part by UTokyo ALPS, JSPS Fellowship (No. 18J13084), JSPS KAKENHI (Nos. 19K03684 and 26220601), JST ERATO (No. JPMJER1601), and MEXT Q-LEAP (No. JPMXS0118068682). D.L.-Q. was an International Research Fellow of JSPS. A.F.v.L. is an International Research Fellow of JSPS.

## Author contributions
S.K. conceived the concept, designed, and fabricated the sample. K.K. and S.K. provided theoretical models. S.K. and D.L.-Q. performed the experiments. D.L.-Q. and S.K. set up the electronics for measurement. S.K. analyzed the results and wrote the manuscript with feedback from Y.N., K.K., A.F.v.L., D.L.-Q., Y.T., and A.N. Y.N. supervised the project.

## Competing interests
The authors declare no competing interests.
