## [Peer Review File · Nature Communications]

REVIEWER COMMENTS

Reviewer #1 (Remarks to the Author):

In their manuscript, "Breaking the trade-off between fast control and long lifetime of a superconducting qubit," the authors demonstrate the operation of a Josephson quantum filter (JQF) and use it to enhance the relaxation time of a superconducting qubit. In contrast to usual Purcell filters that consist of microwave structures that suppress emission at the qubit frequency (while transmitting at the readout frequency), the JQF is a transmon itself that reflects at low powers (a single photon emitted as the result of a qubit relaxation event), yet saturates at higher powers, which include those that are used for qubit control. This allows lifetime enhancement of the qubit without a concomitant increase in drive power for control, which can be detrimental.

The authors present compelling evidence and a rigorous analysis included in a detailed 17 pages of supplemental notes. They use input-output theory to describe waveguide QED, an architecture not as common as the usual circuit QED that is the dominant architecture for building quantum computers with superconducting circuits (indeed that takes place in this work on the readout side with the "usual" Purcell filter), and is also quite different since it treats propagating fields in a waveguide instead of photon modes in a resonator. This is an opportunity to overview waveguide QED, but the essential physics should be a bit more intuitive in the text, and it gets somewhat bogged down by the framework in the supplement. This phenomenon is generic and can be seen (as in Ref. S4, for example) as a diagonalization of the dissipator in Eq. 1 into bright and dark states with different correlated decay rates. Going into the parameter regime where the filter external decay rate is much greater than that of the qubit, the qubit state is approximately the dark state, and is non-radiative. This is all stated in the supplement, but a more thorough explanation should be brought into the main text.

Along the same lines, the physics of JQF saturation should be elaborated on, and how it relates to the work in Ref. 26 and resonance fluorescence. Could the off-resonant split peaks of a Mollow triplet observed in Ref. 26 cause undesired Stark shifts on the qubit?

Overall, the manuscript presents a novel device and thoroughly analyzes its properties, and with the minor clarifications mentioned above, I recommend for publication in Nature Communications.

Reviewer #2 (Remarks to the Author):

The authors have presented a careful, extensive experimental and theoretical study of a device they term the "Josephson Quantum Filter" (JQF). This device acts as an amplitude-sensitive filter for quantum signals, suppressing qubit decay to a control line but allowing strong control signals to pass through to the qubit. In a clever trick, the authors make their JQF frequency-tunable to allow it to be detuned from the qubit, thus isolating the behavior of each device and allowing them to be separately characterized. The authors characterize the JQF itself with direct microwave reflectometry and characterize the qubit with both standard pulsed dispersive readout and with direct reflectometry. They then tune the devices into resonance and then show that the qubit decay is suppressed while Rabi frequencies are constant, i.e. control tones are unaffected by the filter. They present exhaustive analytical and numerical theory in the supplement to support their results. I find no fault with either the experiment or theory, and I regard this as very strong work. I recommend this paper be published after minor revisions.

I have a minor concern about the usefulness of the result which I believe the authors can address

without further experimental work. The device does add a moderate amount of complexity--adding the JQF takes the device from a single transmon and resonator to two of each, with the second transmon flux-tunable. This added complexity must be justified, which the authors do by referencing the reduced power dissipation the JQF allows. This would be a much more compelling statement if the authors can quantitatively cite how much power dissipation is avoided with some reasonable numbers. For example, if one takes 100 qubits, each with a control-line-limited $T_1 = 1$ ms without JQFs, each driven simultaneously with a 10 ns pi-pulse which is 20dB attenuated at the base stage, how much power is dissipated compared with a similar setup with reasonable JQF coupling parameters? How sensitive would this be to fabrication parameters--i.e. will the JQF work with all fixed qubits and resonators assuming state-of-the-art $\sim 1\%$ variation in junction size?

Minor specific comments/suggestions follow:

- 1) The theory relies on the rotating wave approximation. Will this ever break down under reasonable operating conditions? Please add a short discussion to the supplement.
- 2) Would a lumped-element resonator between the qubit and JQF work? Does intrinsic loss in the resonator play a role? Please note this in the main text.
- 3) Does the resonator give any response when the JQF is tuned off resonance? If so please discuss.
- 4) Please discuss the physical significance of the negative correlated decay rate.
- 5) When calculating the single-photon power, is that factor of 4 supposed to be in the denominator or the numerator? Perhaps cite the source for this calculation or re-derive it?
- 6) The qubit temperature with the JQF on resonance is above 90 mK. Is this typical for other qubits without strong control line coupling? Could there be a mechanism by which the JQF causes extra excitations? Please discuss in the main text.
- 7) Typically thermal excitation mechanisms are discussed as part of the overall decay channel. Please justify discussing them as separate channels.
- 8) Please explicitly cite the numbers for T_1 and T_2 enhancement in the main text.
- 9) Please explicitly label the insets in Fig 4, or otherwise indicate that they are zoom-ins.
- 10) Please include a prediction (numerical or analytical) for the maximum coherence enhancement if the resonator length d is exactly half a wavelength.
- 11) Please explain why an incomplete saturation of the JQF causes faster Rabi decay compared to no JQF.
- 12) In the supplement, the JQF dephasing and thermal population are not 0, they are just small enough to be difficult to measure. Please modify the table to indicate this.

Overall, great work, it was very interesting!

-Eli Levenson-Falk

University of Southern California

The authors would like to thank all the reviewers for taking the time to review this manuscript, and for their useful feedback. This document provides responses to every comment made by the reviewers and points out how they are addressed in the revised manuscript if needed.

COMMENTS FROM REVIEWER #1

General comments

Comment 1: *In their manuscript, "Breaking the trade-off between fast control and long lifetime of a superconducting qubit," the authors demonstrate the operation of a Josephson quantum filter (JQF) and use it to enhance the relaxation time of a superconducting qubit. In contrast to usual Purcell filters that consist of microwave structures that suppress emission at the qubit frequency (while transmitting at the readout frequency), the JQF is an transmon itself that reflects at low powers (a single photon emitted as the result of a qubit relaxation event), yet saturates at higher powers, which include those that are used for qubit control. This allows lifetime enhancement of*

the qubit without a concomitant increase in drive power for control, which can be detrimental.

Reply: We thank Reviewer #1 for the detailed description of our work.

Comment 2: *The authors present compelling evidence and a rigorous analysis included in a detailed 17 pages of supplemental notes. They use input-output theory to describe waveguide QED, an architecture not as common as the usual circuit QED that is the dominant architecture for building quantum computers with superconducting circuits (indeed that takes place in this work on the readout side with the "usual" Purcell filter), and is also quite different since it treats propagating fields in a waveguide instead of photon modes in a resonator. This is an opportunity to overview waveguide QED, but the essential physics should be a bit more intuitive in the text, and it gets somewhat bogged down by the framework in the supplement. This phenomenon is generic and can be seen (as in Ref. S4, for example) as a diagonalization of the dissipator in Eq. 1 into bright and dark states with different correlated decay rates. Going into the parameter regime where the filter external decay rate is much greater than that of the qubit, the qubit state is approximately the dark state, and is non-radiative. This is all stated in the supplement, but a more thorough explanation should be brought into the main text.*

Reply: We thank Reviewer #1 for pointing out the issues. Following the suggestion, we added in the main text an explanation of the operation principle of JQF based on the correlated decay rates of bright and dark states.

Modification: Main text, page 3, line 64.

A system composed of a data qubit and a JQF in a semi-infinite control line is described theoretically by the waveguide-quantum-electrodynamics formalism. As shown in Figs. 1c and d, the data qubit is placed at the end of the control line, while the JQF is located a distance d away from the qubit. Here, we consider that the JQF frequency ω_f is set to be identical to the qubit frequency ω_q . The resonant interaction mediated by photons in the control line induces two cooperative effects depending on the distance: a correlated decay and an energy-exchange interaction. In the frame rotating at the qubit and JQF frequencies, the master equation of the composite system of the qubit and JQF with a resonant control field is given by

$$\dot{\hat{\rho}} = -\frac{i}{\hbar} \left[\hat{H}_{\text{eff}} + \hat{H}_{\text{drive}}, \hat{\rho} \right] + \sum_{n,m=q,f} \gamma_{\text{ex}}^{nm} \mathcal{D}(\hat{\sigma}_n, \hat{\sigma}_m) \hat{\rho}, \quad (\text{R1})$$

where $\hat{\sigma}_n$ ($n = q, f$) is the respective lowering operator of the qubit and the JQF, and $\mathcal{D}(\hat{A}, \hat{B})\rho = \hat{B}\hat{\rho}\hat{A}^\dagger - (\hat{A}^\dagger\hat{B}\hat{\rho} + \hat{\rho}\hat{A}^\dagger\hat{B})/2$ is a superoperator describing the correlated decay. The correlated decay terms are described with the individual decay rates of $\gamma_{\text{ex}}^{\text{qq}} = \gamma_{\text{ex}}^{\text{q}}$ and $\gamma_{\text{ex}}^{\text{ff}} = \gamma_{\text{ex}}^{\text{f}} \cos^2(2\pi d/\lambda_{\text{q}})$,

and the correlated decay rates of $\gamma_{\text{ex}}^{\text{qf}} = \gamma_{\text{ex}}^{\text{fq}} = \sqrt{\gamma_{\text{ex}}^{\text{q}}\gamma_{\text{ex}}^{\text{f}}}\cos(2\pi d/\lambda_{\text{q}})$, where γ_{ex}^n ($n = \text{q, f}$) is the respective external coupling rate and λ_{q} is the qubit wavelength. The effective energy-exchange interaction is described as $\hat{H}_{\text{eff}} = \hbar J(\hat{\sigma}_{\text{q}}^{\dagger}\hat{\sigma}_{\text{f}} + \hat{\sigma}_{\text{q}}\hat{\sigma}_{\text{f}}^{\dagger})$, where $J = \sqrt{\gamma_{\text{ex}}^{\text{q}}\gamma_{\text{ex}}^{\text{f}}}\sin(2\pi d/\lambda_{\text{q}})/2$ is the coupling strength. Moreover, the drive Hamiltonian is given by $\hat{H}_{\text{drive}} = \hbar\Omega_{\text{q}}\hat{\sigma}_x^{\text{q}}/2 + \hbar\Omega_{\text{f}}\cos(2\pi d/\lambda_{\text{q}})\hat{\sigma}_x^{\text{f}}/2$, where $\Omega_n = 2\sqrt{\gamma_{\text{ex}}^n\dot{n}}$ is the Rabi frequency with a control photon flux of \dot{n} and $\hat{\sigma}_s^n$ is the Pauli s operator ($s = x, y, z$).

Here, we explain the working principle of the JQF based on the master equation (R1). The seemingly contradictory goal is that the qubit is isolated from the JQF while employing the sub-radiance effect protecting the qubit from decaying. We achieve this by preparing a system where the qubit and the JQF are coupled with the strongly asymmetric external coupling rates to the control line, as follows. To avoid the qubit hybridizing with the JQF ($J = 0$) and to maximize the correlated decay ($|\gamma_{\text{ex}}^{\text{qf}}| = |\gamma_{\text{ex}}^{\text{fq}}| = \sqrt{\gamma_{\text{ex}}^{\text{q}}\gamma_{\text{ex}}^{\text{f}}}$), the distance d is set to half the qubit wavelength ($d = \lambda_{\text{q}}/2$). Therefore, the correlated decays are described as the individual decays of a bright mode [$\hat{\sigma}_{\text{B}} = \mathcal{N}(-\sqrt{\gamma_{\text{ex}}^{\text{q}}}\hat{\sigma}_{\text{q}} + \sqrt{\gamma_{\text{ex}}^{\text{f}}}\hat{\sigma}_{\text{f}})$] with decay rate $\gamma_{\text{ex}}^{\text{B}} = \gamma_{\text{ex}}^{\text{q}} + \gamma_{\text{ex}}^{\text{f}}$ and a dark mode [$\hat{\sigma}_{\text{D}} = \mathcal{N}(\sqrt{\gamma_{\text{ex}}^{\text{f}}}\hat{\sigma}_{\text{q}} + \sqrt{\gamma_{\text{ex}}^{\text{q}}}\hat{\sigma}_{\text{f}})$] with decay rate $\gamma_{\text{ex}}^{\text{D}} = 0$, where \mathcal{N} is the normalization factor. By engineering the system such that $\gamma_{\text{ex}}^{\text{f}} \gg \gamma_{\text{ex}}^{\text{q}}$, the excited state of the qubit is close to the dark state ($\hat{\sigma}_{\text{q}} \approx \hat{\sigma}_{\text{D}}$), suppressing its radiative decay. In the limit of $\gamma_{\text{ex}}^{\text{f}} \gg \gamma_{\text{ex}}^{\text{q}}$, the master equation (R1) can be approximated as

$$\dot{\hat{\rho}} = -\frac{i}{\hbar}[\hat{H}_{\text{drive}}, \hat{\rho}] + \mathcal{D}\left(\sqrt{\gamma_{\text{ex}}^{\text{f}}}\hat{\sigma}_{\text{f}} - \sqrt{\gamma_{\text{ex}}^{\text{q}}}(1 + \hat{\sigma}_z^{\text{f}})\hat{\sigma}_{\text{q}}\right)\hat{\rho}, \quad (\text{R2})$$

with the effective drive Hamiltonian originating from the correlated decay:

$$\hat{H}_{\text{drive}} = -\frac{\hbar\Omega_{\text{f}}}{2}\hat{\sigma}_x^{\text{f}} + \frac{\hbar\Omega_{\text{q}}}{2}(1 + \hat{\sigma}_z^{\text{f}})\hat{\sigma}_x^{\text{q}}, \quad (\text{R3})$$

where $\mathcal{D}(\hat{A})\hat{\rho} = \hat{A}\hat{\rho}\hat{A}^{\dagger} - (\hat{A}^{\dagger}\hat{A}\hat{\rho} + \hat{\rho}\hat{A}^{\dagger}\hat{A})/2$. From this approximative master equation, we find that the qubit is coupled to the control line depending on the state of the JQF. When the control field is absent ($\hat{H}_{\text{drive}} = 0$), the JQF is in the ground state ($\hat{\sigma}_z^{\text{f}} = -1$), resulting in the complete suppression of the decay term of the qubit. When the control field is applied, on the other hand, the conditional drive term of the drive Hamiltonian ($\propto \hat{\sigma}_z^{\text{f}}\hat{\sigma}_x^{\text{q}}$) is suppressed by the drive term of the JQF ($\propto \hat{\sigma}_x^{\text{f}}$) in the secular approximation, enabling us to individually drive the qubit with the Rabi frequency of Ω_{q} (see more details in Supplementary Note 4).

Comment 3: *Along the same lines, the physics of JQF saturation should be elaborated on, and how it relates to the work in Ref. 26 and resonance fluorescence. Could the off-resonant split peaks*

of a Mollow triplet observed in Ref. 26 cause undesired Stark shifts on the qubit?

Reply: We think that the Mollow triplet signals from the driven JQF do not induce any significant ac Stark shift of the qubit because of the following reasons: The photon flux of the resonance fluorescence from the JQF is on the order of $\gamma_{\text{ex}}^{\text{f}}$, which corresponds to the qubit drive amplitude on the order of $\sqrt{\gamma_{\text{ex}}^{\text{q}}\gamma_{\text{ex}}^{\text{f}}}$. The detuning of the sideband fluorescence with respect to the qubit is the Rabi frequency of the JQF, i.e. Ω_{f} . Thus, the ac Stark shift of the qubit induced by the fluorescence can be estimated to be on the order of $\gamma_{\text{ex}}^{\text{q}}\gamma_{\text{ex}}^{\text{f}}/\Omega_{\text{f}}$. In the saturation regime of the JQF ($\Omega_{\text{f}}/\gamma_{\text{ex}}^{\text{f}} \gg 1$), the Stark shift should be much smaller than the Rabi frequency of the qubit, Ω_{q} . The sidebands of the Mollow triplet also appear symmetrically on both sides of the qubit frequency, which would cancel the ac Stark shift as well.

Comment 4: *Overall, the manuscript presents a novel device and thoroughly analyzes its properties, and with the minor clarifications mentioned above, I recommend for publication in Nature Communications.*

Reply: We are very grateful to Reviewer #1 for the positive comment and for recommending our work for publication in Nature Communications.

COMMENTS FROM REVIEWER #2

General comments

Comment 1: *The authors have presented a careful, extensive experimental and theoretical study of a device they term the “Josephson Quantum Filter” (JQF). This device acts as an amplitude-sensitive filter for quantum signals, suppressing qubit decay to a control line but allowing strong control signals to pass through to the qubit. In a clever trick, the authors make their JQF frequency-tunable to allow it to be detuned from the qubit, thus isolating the behavior of each device and allowing them to be separately characterized. The authors characterize the JQF itself with direct microwave reflectometry and characterize the qubit with both standard pulsed dispersive readout and with direct reflectometry. They then tune the devices into resonance and then show that the qubit decay is suppressed while Rabi frequencies are constant, i.e. control tones are unaffected by the filter. They present exhaustive analytical and numerical theory in the supplement to support their results. I find no fault with either the experiment or theory, and I regard this as very strong*

work. I recommend this paper be published after minor revisions.

Reply: We thank Reviewer #2 for the careful and positive evaluation of our work and for the recommendation for publication in Nature Communications.

Comment 2: *I have a minor concern about the usefulness of the result which I believe the authors can address without further experimental work. The device does add a moderate amount of complexity—adding the JQF takes the device from a single transmon and resonator to two of each, with the second transmon flux-tunable.*

Reply: Regarding the additional complexity, our proposal is to add an ancillary qubit acting as a JQF on the control line of a data qubit at an appropriate distance apart. As discussed in Reply to comment 4 made by Reviewer #2, the JQF works properly within its bandwidth which corresponds to about 1% of the qubit frequency. With an advanced fabrication process providing good junction homogeneity, the frequency-tunable JQF can be replaced with a frequency-fixed JQF, which, we believe, minimizes the additional complexity.

Comment 3: *This added complexity must be justified, which the authors do by referencing the reduced power dissipation the JQF allows. This would be a much more compelling statement if the authors can quantitatively cite how much power dissipation is avoided with some reasonable numbers. For example, if one takes 100 qubits, each with a control-line-limited $T_1 = 1$ ms without JQFs, each driven simultaneously with a 10 ns pi-pulse which is 20dB attenuated at the base stage, how much power is dissipated compared with a similar setup with reasonable JQF coupling parameters?*

Reply: In a conventional setup, a superconducting qubit with a state-of-the-art relaxation time (≈ 1 ms) should be coupled to its control line with the external coupling rate on the order of 100 Hz or less. On the other hand, the external coupling rate of a qubit accompanied by a JQF on its control line can be set to the order of 100 kHz, allowing 0.1% gate error at a 100-MHz Rabi frequency. In this example, the control power required for a given Rabi frequency of the qubit is reduced by a factor of 30 dB compared to the conventional control setup.

Let us consider a setup where 100 qubits are controlled at 100-MHz Rabi frequencies by microwave fields 20-dB attenuated at the base-temperature (10 mK) stage. While the total control power of -20 dBm is dissipated at the 10-mK stage in a setup without JQFs, only -50 dBm is dissipated in a setup with JQFs. The power consumption of -20 dBm is already comparable to the usual cooling power of a dilution refrigerator at 10 mK (≈ 10 μ W). Therefore, the scheme with

JQFs would give a large impact on the heat management in the control lines.

We however refrain from putting in the main text this specific value of power consumption based on the hand-waving argument. In the real setup, there could be many other factors to be considered. Instead, we mentioned the ratio of the control power we can save using the JQF compared to a conventional control setup.

Modification: Main text, page 7, line 203.

We experimentally confirmed that the JQF suppresses the qubit radiative decay, while it does not significantly reduce the Rabi frequency and the gate fidelity of the qubit from those without the JQF. This would also allow us to enhance the qubit–control-line coupling, and thus to reduce the Rabi drive power, by three orders of magnitude, compared with the weak coupling required for a state-of-the-art qubit with a 1-ms lifetime in the absence of JQF.

Comment 4: *How sensitive would this be to fabrication parameters—i.e. will the JQF work with all fixed qubits and resonators assuming state-of-the-art 1% variation in junction size?*

Reply: As Fig. 3(b) in the main text shows, the coherence enhancement spectrum of the qubit is observed within the JQF bandwidth. In our experiment, the JQF bandwidth is about $\gamma_{\text{ex}}^f/2\pi \approx 100$ MHz, which is about 1% of the JQF frequency. Therefore, we believe that a fixed-frequency JQF can be implemented using a fabrication process with the state-of-the-art Josephson-energy variation. We added a sentence addressing this aspect in the main text.

Modification: Main text, page 5, line 148.

Since this bandwidth corresponds to about 1% of the JQF frequency, it would be possible to implement a frequency-fixed JQF in a future device with the state-of-art Josephson-energy variation^{R1}.

Specific comments

Comment 5: *1) The theory relies on the rotating wave approximation. Will this ever break down under reasonable operating conditions? Please add a short discussion to the supplement.*

Reply: We think that the rotating wave approximation should be valid as long as the resonance frequencies of the qubit and JQF are much larger than their decay rates.

Modification: Supplementary Information, page 3, Note 3.

Note that the rotating wave approximation should be valid as long as the resonance frequencies of

the qubit and JQF are much larger than their external coupling rates.

Comment 6: 2) *Would a lumped-element resonator between the qubit and JQF work? Does intrinsic loss in the resonator play a role? Please note this in the main text.* 3) *Does the resonator give any response when the JQF is tuned off resonance? If so please discuss.*

Reply: In our setup, both the data qubit and the JQF are coupled directly to a control line, and they interact with each other via propagating photons in the control line and form the dark and bright states. The structure proposed by Reviewer #2 requires a totally different consideration and is out of scope of our approach. As first glance, it is not likely that a lumped-element (or distributed-element) resonator would do the same job.

Comment 7: 4) *Please discuss the physical significance of the negative correlated decay rate.*

Reply: The sign of the correlated decay rate, which is determined from the JQF–qubit distance, is not crucial for the formation of the dark mode enabling the JQF to suppress the qubit decay. The sign corresponds only to the phase of the superposition of the dark mode, and has no further physical significance.

Comment 8: 5) *When calculating the single-photon power, is that factor of 4 supposed to be in the denominator or the numerator? Perhaps cite the source for this calculation or re-derive it?*

Reply: The factor of 4 is in the denominator. We modified the text and added a reference.

Modification: Main text, page 13, caption in Fig. 2.

The probe power of -120 dBm corresponds to the single-photon power level for the JQF, defined as $\hbar\omega_f(\gamma_{\text{ex}}^f + \gamma_{\text{in}}^f)^2 / (4\gamma_{\text{ex}}^f)$, which would populate a linear resonator with a single photon on average^{R2}.

Comment 9: 6) *The qubit temperature with the JQF on resonance is above 90 mK. Is this typical for other qubits without strong control line coupling? Could there be a mechanism by which the JQF causes extra excitations? Please discuss in the main text.*

Reply: In our setup, the typical effective temperature of the qubit is about 100 mK, regardless of circuit configuration. The dominant factor of the relatively high effective temperature can be a thermal noise from the intrinsic loss channel, such as radiation fields from the input and output lines in the infrared frequency range or from the external environment of the sample package. Thus, we don't think that adding the ancillary transmon qubit acting as the JQF increases the effective temperature of the data qubit. We added a more specific explanation in the main text

about the increase in the qubit temperature when the JQF is on resonance.

Modification: Main text, page 5, line 139.

The thermal population of the qubit is increased from 2.8% to 16.2%, which is not because the JQF adds a thermal noise to the qubit, but because it decouples the qubit from the control line which has a lower effective temperature than the intrinsic loss channel.

Comment 10: 7) *Typically thermal excitation mechanisms are discussed as part of the overall decay channel. Please justify discussing them as separate channels.*

Reply: In the manuscript, we consider an effective model where the qubit is coupled to two different decay channels, i.e. the external coupling and intrinsic loss channels. The external coupling rate is defined as the radiative decay rate into the control line, while the intrinsic loss rate is defined as the total decay rate to all other decay paths including radiation to other modes and nonradiative ones. These two channels are considered to have individual temperatures in this model. Note that the temperature of the intrinsic loss channel is effectively determined from all the contributions within the channel. We apply the same treatment to the JQF.

Comment 11: 8) *Please explicitly cite the numbers for T_1 and T_2 enhancement in the main text.*

Reply: We indicated the explicit numbers of the improvements in T_1 and T_2^E in the main text.

Modification: Main text, page 5, line 143.

Both the relaxation and coherence times are enhanced by a factor of about 4 when the JQF is nearly on resonance with the qubit.

Comment 12: 9) *Please explicitly label the insets in Fig 4, or otherwise indicate that they are zoom-ins.*

Reply: We added a sentence explaining the insets in the caption of Fig.4.

Modification: Main text, page 15, caption in Fig. 4.

The insets are magnifications of the small amplitude region of the main plots.

Comment 13: 10) *Please include a prediction (numerical or analytical) for the maximum coherence enhancement if the resonator length d is exactly half a wavelength.*

Reply: Even if the JQF is located at a distance not exactly half a wavelength apart, with an appropriate JQF–qubit detuning we are able to enhance the qubit coherence to the level that

would be achieved with the ideal distance. In Fig. R1, a numerically simulated result for the non-ideal distance ($d = 0.526\lambda_q$) is compared with that for the ideal distance ($d = 0.5\lambda_q$). Note that the same system parameters are used in these numerical simulations except for the distance. We conclude that the requirement of the ideal distance is not strict as long as the JQF–qubit detuning is adjusted properly.

Modification: Main text, page 6, line 152.

Interestingly, for this distance, we find in numerical simulations that with a finite detuning of $(\omega_f - \omega_q)/2\pi = 9$ MHz the decay times of the qubit can reach the level that would be achieved with the ideal distance of $d = 0.5\lambda_q$ (see Supplementary Note 9).

Modification: Supplementary Information, page 16, Note 9.

The dotted lines in Figs. S9b and f are the decay times numerically calculated with the ideal distance $d = 0.5\lambda_q$. Note that the same system parameters are used in these numerical simulations except for the distance. Both the results with $d = 0.526\lambda_q$ and $d = 0.5\lambda_q$ show the equivalent enhancements in the decay times at the different optimal JQF–qubit detuning. In other words, even if the JQF is located at a distance not exactly half a wavelength apart, with an appropriate JQF–qubit detuning we are able to enhance the qubit coherence to the level that would be achieved with the ideal distance.

Modification: Supplementary Information, page 16, caption in Fig. S9.

The circles and solid lines are the experimental results and the theoretical fits with the actual distance $d = 0.526\lambda_q$, respectively. The dotted line is the numerical simulation calculated with the ideal distance $d = 0.5\lambda_q$. [...] The dotted lines are the corresponding numerical calculations with the ideal distance $d = 0.5\lambda_q$.

Modification: Supplementary Information, page 16, Figs. S9b and f.

Comment 14: *11) Please explain why an incomplete saturation of the JQF causes faster Rabi decay compared to no JQF.*

Reply: It is explained by the waveguide QED formalism, which reproduces the experimental data well. We modified the explanation of the working principle of the JQF in the main text according to comment 2 made by Reviewer #1. When the qubit is driven via the control waveguide, an unwanted interaction between the qubit and JQF ($\propto \hat{\sigma}_x^q \hat{\sigma}_z^f$), which originates from the correlated decay, is induced. Although it is suppressed by the JQF drive ($\propto \hat{\sigma}_x^f$) in the secular approximation, this interaction inevitably provides additional decays of the qubit via the JQF. Thus, when the ground and first excited states of the transmon JQF are populated ($\Omega_f/\gamma_{\text{ex}}^f \approx 1$), the Rabi decay

FIG. R1. Relaxation time of the qubit as a function of the JQF-qubit detuning. The circles are the experimental results, while the orange and blue solid lines are the numerically simulated results with the distances $d = 0.526\lambda_q$ and $d = 0.5\lambda_q$, respectively.

rate of the qubit is increased due to this unwanted interaction. When the JQF is excited to its higher levels with a strong drive ($\Omega_f/\gamma_{\text{ex}}^f \gg 1$), there is no correlated decay between the qubit and the higher levels of the JQF due to the large detuning. Therefore, the qubit Rabi oscillation is not affected by the transmon JQF in the strong drive regime (see more details in Supplementary Note 4).

Modification: Main text, page 7, line 182.

This is because the saturated two-level JQF is still coupled to the qubit [see the conditional drive term originating from the correlated decay in Eq. (3)], providing additional decay channels of the qubit through the JQF. The transmon JQF, on the other hand, once excited to its higher levels, becomes decoupled from the qubit due to its anharmonicity and loses the correlated decay, and therefore no longer affects the Rabi oscillations of the qubit.

Comment 15: *12) In the supplement, the JQF dephasing and thermal population are not 0, they are just small enough to be difficult to measure. Please modify the table to indicate this.*

Reply: As Reviewer #2 mentioned, the dephasing rate and the thermal quanta of the JQF cannot be zero in the real device. These two values are found to be too small to be reliably characterized by fitting the experimental data of the JQF reflection spectra. We modified the table accordingly.

Modification: Supplementary Information, page 11, Note 5

From these fits, the system parameters of the JQF are found to be $\omega_f/2\pi = 8.0004$ GHz,

$\gamma_{\text{ex}}^f/2\pi = 112$ MHz, and $\gamma_{\text{in}}^f/2\pi = 3$ MHz. The pure dephasing rate γ_{ϕ}^f and the intrinsic thermal quanta n_{th}^f are found to be too small to be reliably characterized by fitting the reflection spectra of the JQF over-coupled to the control line. We only estimate the approximate upper bounds for them by numerical simulations (Table S1).

Modification: Supplementary Information, page 2, Table S1.

Pure dephasing rate, $\gamma_{\phi}^f/2\pi$	$\lesssim 100$ kHz
Intrinsic thermal quanta, n_{th}^f	≈ 0

- [R1] J. M. Kreikebaum, K. P. O'Brien, A. Morvan, and I. Siddiqi, Superconductor Science and Technology **33**, 06LT02 (2020).
- [R2] A. A. Clerk, M. H. Devoret, S. M. Girvin, F. Marquardt, and R. J. Schoelkopf, Rev. Mod. Phys. **82**, 1155 (2010).

REVIEWERS' COMMENTS:

Reviewer #1 (Remarks to the Author):

I thank the authors for their detailed response to my comments and the modification of their manuscript. It is now suitable for publication in Nature Communications.

Reviewer #2 (Remarks to the Author):

The authors have answered all my concerns. I recommend publication.